# The Use of Prefemoral Endoscope-Assisted Surgery and Transplastron Coeliotomy in Chelonian Reproductive Disorders

**DOI:** 10.3390/ani12233439

**Published:** 2022-12-06

**Authors:** Tom Hellebuyck, Ferran Solanes Vilanova

**Affiliations:** Department of Pathobiology, Pharmacology and Zoological Medicine, Faculty of Veterinary Medicine, Ghent University, Salisburylaan 133, B-9820 Merelbeke, Belgium

**Keywords:** chelonians, coeliotomy, endoscopy, oophorectomy, transplastronectomy

## Abstract

**Simple Summary:**

Tortoises and turtles (chelonians) are routinely presented in veterinary practice because of their reproductive disorders. Although egg binding in chelonians can often be resolved with conventional therapy, the diagnosis and treatment of complicated cases of egg binding and various other disorders of the reproductive tract in chelonians often require a more advanced therapeutic approach. If surgical intervention is required, endoscope-assisted techniques comprise the least invasive and thus preferred surgical approach. In cases where the use of endoscope-assisted procedures is not feasible, the reproductive tract needs to be accessed through the plastron (transplastron coeliotomy). The present report describes the diagnostic and surgical approach applied in seven cases of female chelonians with reproductive disorders. The therapeutic efficacy largely relies on the choice of minimally invasive endoscope-assisted surgery versus transplastron coeliotomy.

**Abstract:**

Throughout the last decades, the increased popularity of the keeping of reptiles has led to a better understanding of the captive needs of a wide variety of species. Although this is reflected by the successful captive reproduction in many of those species, reproductive disorders such as preovulatory follicular stasis, postovulatory dystocia, secondary yolk coelomitis, and prolapse of the oviduct and male copulatory organ are commonly encountered in veterinary practice. In comparison to squamates, chelonians with postovulatory dystocia seem to be more responsive to oxytocin treatment, even in cases of chronic dystocia. There are various conditions, however, that necessitate the use of surgical procedures for the treatment of dystocia and other reproductive disorders in chelonians. Although restrictions may be encountered, the endoscope-assisted prefemoral approach is the least invasive and thus preferred technique instead of the ventral transplastron coeliotomy. The present report describes the diagnostic and surgical approach applied in seven cases of female chelonians with reproductive disorders. The therapeutic efficacy largely relied on the choice of minimally invasive endoscope-assisted surgery versus transplastron coeliotomy that was primarily dictated by the involved species, etiology, and associated pathology.

## 1. Introduction

The increased popularity of reptile pets has led to continued advances in husbandry and nutrition and the successful breeding of a wide variety of species. Nevertheless, reproductive disorders such as preovulatory follicular stasis (PFS) and dystocia are still some of the most frequently observed disorders in the reptile patient [1,2]. Follicular activity and ovulation can occur in solitary female reptiles, mostly resulting in the production of infertile ova. Although PFS and dystocia are mostly seasonal problems, environmental control in captivity may result in reproductive activity throughout the year, especially in nontemperate species [1,2,3].

Dystocia can be classified into obstructive or nonobstructive dystocia. Obstructive dystocia is caused by a barrier that prevents the normal passage of eggs or feti through the oviduct or cloaca and may be related to a maternal or fetal abnormality including deformation of the pelvis, space-occupying processes in the coelomic cavity, oviductal strictures, adhesive salpingitis and eggs or feti with an abnormal shape, size, or mineralization as well as broken or damaged eggs [2]. Nonobstructive dystocia can be attributed to a wide variety of primary etiologies, including inappropriate environmental factors and malnutrition, mainly those that result in calcium deficiency, or any disorder that results in a poor physical condition of the female. The etiology is mostly multifactorial, and often it is challenging to obtain a proven causation [1,2].

True nonobstructive dystocia is routinely treated with the administration of oxytocin or, if available, arginine vasotocin, but therapeutic efficacy is variable and seems to be higher in chelonians than in squamates [4,5]. If medical treatment fails or if a permanent solution is desirable, surgical intervention is required. In squamates, reproductive surgery, mainly consisting of oophorectomy, salpingotomy, and/or salpingectomy, is routinely and relatively easily performed through coeliotomy. Surgical intervention for the treatment of dystocia and other reproductive disorders in chelonians, on the other hand, is more challenging due to the constraints of the chelonian shell [6,7]. Whenever possible, the minimally invasive endoscope-assisted prefemoral approach is the preferred technique instead of the ventral transplastron coeliotomy. The latter is largely dictated by the anatomical characteristics of the involved species, and especially in cases of obstructive dystocia, the primary and secondary pathologies [6,7,8,9].

The present report describes the diagnostic and therapeutic approach of reproductive disorders in seven cases of female chelonians with an emphasis on the deciding factors towards selecting a transplastron coeliotomy or endoscope-assisted surgical approach.

## 2. Materials and Methods

### 2.1. Animals and Diagnostic Procedures

The cases in this study involve female chelonians that were presented at a veterinary teaching hospital with signs that were attributed to a reproductive disorder. Table 1 provides an overview of the species, the age, and bodyweight at initial presentation for each case, as well as the provided housing and feeding regimen.

If a radiographic examination was performed, radiographic projections comprised a laterolateral and craniocaudal projection (horizontal beam) as well as a dorsoventral projection (vertical beam). An ultrasonographic examination was performed using an 8–15 MHz microconvex transducer using the prefemoral coupling window. Unless stated otherwise, ultrasonographic abnormalities were confined to the reproductive tract.

### 2.2. Anesthetic Procedures

In all cases, an identical anesthetic and analgesic protocol was used consisting of intravenous (IV) induction of anesthesia with alfaxalone (10 mg/kg, Alfaxan Multidose, Jurox Limited, Crawley, UK) administered via the right jugular vein and intubation with an uncuffed endotracheal tube. Anesthesia was maintained with 1.5–2% isoflurane (Isoflo, Abbott Logistics B.V., Breda, The Netherlands) in 1 L medical oxygen per minute with intermittent positive-pressure ventilation. In case of performing a transplastron coeliotomy, morphine was administered perioperatively at 2 mg/kg (IM, Morphine HCL Sterop 10 mg/mL, Laboratoires Sterop NV, Brussels, Belgium) and tramadol (5 mg/kg, PO, Tramadol EG, Eurogenerics NV, Brussels, Belgium) postoperatively. For the endoscope-assisted prefemoral approach, local infiltration with 2 mg/kg lidocaine (Xylocaine 1%, Aspen Pharma Trading Limited, Dublin, Ireland) of the prefemoral incision site was performed prior to the start of the surgery, and meloxicam was administered perioperatively (0.3–0.5 mg/kg, IM, Metacam 20 mg/mL, Boehringer Ingelheim, Vetmedica GmbH, Ingelheim, Germany).

### 2.3. Surgical Procedures

Routine considerations for preanesthetic evaluation and surgical preparation were performed, and fasting times varied between 24 and 36 h. The chelonians were encouraged to defecate and urinate before surgery by stimulation of the cloaca using a cotton tip applicator or shallow bathing in the turtles and testudinid species, respectively.

For the endoscope-assisted prefemoral procedures, a routine approach as previously described by Innes and Hernandez-Divers [10] and Proença and Divers [11] was used to enter the coelomic cavity with the animal placed in lateral recumbency. Depending on the species and/or the surgical aim, a left or bilateral prefemoral approach was applied.

Transplastron coeliotomies were performed as described by Divers and Wüst [6] and special preoperative considerations included assessing the plastron thickness, including its expansion towards the plastrocarpacial bridge, and locating the hinge region and the degree of access required to perform the procedure. An oscillating sagittal saw was used to incise the plastron, and the caudal cut was generally made incomplete, leaving a few millimeters of bone thickness and allowing the segment to break spontaneously when lifting the segment with a periostal elevator. The latter was considered to help stabilizing the loose segment during closure and increased the chance of maintaining blood supply and thus primary healing. Entry into the coelomic cavity was achieved by making a midline incision of the coelomic membrane between the abdominal veins in the Testudo species and a unilateral paramedian incision in the (semi-)aquatic species. For closure of the plastron incision, veterinary acrylic (Technovit 6091, Kulzer GmbH, Hanau, Germany) was used.

## 3. Results

### 3.1. Case 1

A 29-year-old female red-eared slider (*Trachemys scripta elegans*) was presented with anorexia and apathy after 2 weeks. During the past 22 years, the turtle annually produced 4 to 6 eggs and had no clinical history. Radiographic projections (Figure 1A,B) revealed the presence of 5 eggs located at the right side of the caudal coelomic cavity. Three eggs had an irregular shape, and one egg had an unusual small size and seemed to be fused with one of the other eggs. All eggs showed abnormally thickened shells, and a lamellar appearance was noticed in the most cranial egg. A generalized soft tissue opacity occupying the entire coelomic cavity was noted, causing obvious compression of the lung field on the laterolateral (Figure 1B) and craniocaudal projection. An ultrasound confirmed the presence of retained eggs and the presence of a large soft tissue mass measuring approximately 11.5 × 6.1 cm (Figure 1C) at the left-mid- to caudal coelomic cavity. The mass showed a heterogenic appearance, and a color doppler examination revealed pronounced vascularization. The latter findings yielded obstructive dystocia and chronic egg retention caused by a space-occupying mass.

Taking into account the size of the mass and the presence of multiple retained eggs with an abnormal appearance, the transplastron coeliotomy was chosen as the most appropriate surgical procedure for this case. The mass appeared to originate from the right ovary. Following ligation and resection of the mass, static displacement of the heart, liver, gastrointestinal tract, and left oviduct to the right coelomic cavity was noted. Based on a visual inspection, no signs of metastasis could be observed. Next, a bilateral salpingotomy was performed to remove two eggs from each oviduct followed by the oophorectomy of the right ovary that showed an inactive and normal appearance. Recovery from anesthesia was uneventful, and the turtle showed a good appetite and normal behavior the day after surgery. The histological examination of the ovarian mass revealed a cell-rich tissue that was growing infiltrative in dense collagen stroma (Figure 1D). The neoplastic cells were organized in islets to multi-layered tubules and showed a moderate amount of clear, polygonal cytoplasm. The nuclei were pale with a small nucleolus, and the number of mitosis per high power field was less than one. A histological diagnosis of a scirrhous ovarian carcinoma was made. During a 2-year follow-up period, the turtle showed a good general condition, and based on the ultrasonography and radiographic examination performed 6 and 18 months after surgery, no indications of neoplastic disease could be noted.

### 3.2. Case 2

A 19-year-old female yellow-bellied slider (*Trachemys scripta scripta*) was presented because of anorexia and having displayed overactive behavior for two weeks. During the past 10 years, the turtle had produced two clutches of eggs per year with, on average, 8 eggs per clutch. During the past 5 years, recurrent episodes of dystocia were resolved following the administration of oxytocin by a local vet. Two days before initial presentation, the turtle had received 3 administrations of oxytocin (IM, Oxytocine Kela 10 IU/mL, Kela Veterinaria NV, Antwerpen, Belgium) at 15 IU/kg without effect. An ultrasound revealed the presence of at least 2 oval-shaped eggs, and a radiographic examination (Figure 2A,B) confirmed the presence of 4 eggs with a well-mineralized but thin shell. Three eggs had an abnormal shape, and one egg showed an abnormally small size. The most caudal, large egg showed a vertical position and was located cranial to the pelvis inlet.

Based on these findings, and especially because of the positioning of the most caudal egg, obstructive dystocia was presumed, and an endoscope-assisted left coeliotomy was performed. All eggs appeared to be free in the coelomic cavity. Despite the large size of the eggs relative to the size of the prefemoral incision, an endoscope-assisted localization and manipulation of the eggs followed by bilateral oophorectomy was attempted instead of performing a transplastron coeliotomy. Taking into account the diameter of the eggs in relation to the size of the prefemoral incision, ovocentesis needed to be performed using a 19 gauge needle. After aspiration of the content of the eggs, the eggshells were collapsed and exteriorized. Once the eggs were removed, a bilateral oophorectomy was performed via the left prefemoral fossa. No indications of oviductal rupture were found. Recovery was uneventful, and one week postoperatively the turtle displayed a normal and active behavior, and the appetite was restored. The turtle remained healthy during a 4-year follow-up period.

### 3.3. Case 3

A 32-year-old female common snapping turtle (*Chelydra serpentina*) was presented because of post-hibernation anorexia. The turtle was housed together with a male of the same age in an outside pond where the animals hibernated from early September to late April. Although mating behavior had been noticed, the female never produced eggs.

A radiographic examination did not reveal abnormalities, but based on the ultrasound, both ovaries contained previtellogenic follicles as well as a large number of heterogeneous vittelogenic follicles with an average diameter of 1.2 cm showing an anechogenic central core and a peripheral anechoic rim. Blood was collected from the jugular vein for a serum biochemistry profile and hematological evaluation. Besides hypercalcemia (14.9 mg/dL), the serum biochemistry did not reveal abnormalities in comparison to physiological reference intervals [12], and a complete blood cell count revealed pronounced heterophilia and monocytosis. Based on these findings, a presumptive diagnosis of oophoritis was made, and an endoscope-assisted oophorectomy was planned, taking into account the typically nonrestrictive prefemoral fossa in this mature snapping turtle. Following a routine left prefemoral approach to the coelom, the ipsilateral ovary was located, and a 3 mm atraumatic forceps was used to grasp the interfollicular tissue and retract the ovary to the prefemoral incision. Next, the whole ovary was gently exteriorized and the mesovarium dissected using radiosurgery. Following the same procedure, the contralateral ovary was removed.

The macroscopic appearance of both ovaries (Figure 3A,B) complied to the ultrasonographic findings with adhesions of multiple follicles to the coelomic wall, and a bilateral oophorectomy was performed. A moderate amount of free coelomic fluid with a blurry appearance and diphtheroid plaques at the serosal surface of the liver were noticed. As the snapping turtle showed persistent anorexia following surgery, force-feeding was performed every 3 days. A histological examination of ovarian tissue revealed the presence of a small number of previtellogenic follicles and multiple vitellogenic follicles that contained protein globules surrounded by large numbers of foamy macrophages and giant cells (Figure 3C). Occasionally, aggregates of lymphocytes, plasma cells, and heterophils were noticed. In the ovarian parenchyma, the multifocal presence of cholesterol crystals that were surrounded by macrophages and giant cells could be observed (Figure 3D). Ziehl Neelsen staining was negative. A histological diagnosis of granulomatous inflammation with the formation of cholesterol granulomas, presumably caused as a reaction to the leakage of yolk, was made. A microbiological examination of ovarian tissue yielded negative results. Four weeks postoperatively, the turtle regained her appetite and was sent home. During the first months of an 18-month follow-up period, the owner stated that the appetite largely exceeded the appetite that was seen preoperatively, and 6 months postoperatively the bodyweight had increased to 12.3 kg. After adapting the feeding schedule, the body weight was reduced to the preoperative body weight 3 months later and remained stable until the end of the 18 month follow-up period.

### 3.4. Case 4

A 19-year-old female black marsh turtle (*Siebenrockiella crassicollis*) was presented with pronounced apathy and anorexia after one week. During the previous year, the turtle had produced 2 to 4 fertilized eggs each year. Based on a radiographic examination, a mild but decreased mineralization of the skeleton and a large, excessively mineralized egg were seen in the left caudal coelomic cavity, while in the right caudal coelomic cavity, the remnants of an eggshell could be noticed (Figure 4A,B). An ultrasound confirmed the presence of a well-developed egg in the left oviduct and a hyperechoic structure surrounded by a moderate amount of mildly hyperechoic fluid in the right oviduct. As the turtle did not respond to 3 injections with oxytocin at increasing doses of 5, 10, and 15 IU/kg administered at 2 h intervals, an endoscope-assisted bilateral salpingotomy was planned. The prefemoral fossa was deemed nonrestrictive, and it was considered that both oviducts could be reached using a unilateral approach. Via a routine left prefemoral approach to the coelom, the ipsilateral oviduct was identified and grasped using a 3 mm atraumatic forceps, and a left salpingotomy was performed. Obvious inflammation of the oviductal mucosa was noticed, and a decayed and collapsed eggshell that contained thickened yolk as well as free fluid was removed from the oviduct and sampled for microbiological examination. Next, the contralateral oviduct was located and exteriorized via the left prefemoral incision, and the egg was removed through salpingotomy. Postoperatively, antimicrobial treatment with amoxicillin-clavulanic acid (20 mg/kg, PO, once daily, 10 days, Synulox 50 mg, Zoetis Belgium SA, Louvain-la-Neuve, Belgium) was initiated, and the appetite was restored after four days. A microbiological examination revealed a pure culture of *Citrobacter freundii* that was sensitive to the used antimicrobial treatment. The bacterium was presumed to have established salpingitis following ascending infection, secondary to egg retention and decay. During a one-year follow-up period, the turtle did not show recurrence of salpingitis. Despite several mating attempts, the turtle did not ovulate or produce eggs during this period.

### 3.5. Case 5

A 22-year-old female red-eared slider was presented with buoyancy disorder and tenesmus after one week. During the past decade, the turtle produced a clutch of 6 to 8 eggs each year. A radiographic examination revealed the presence of 7 eggs with a well-mineralized shell, but the two most cranial eggs with a normal size as well as an excessively small egg were collapsed. The most caudal egg had a relatively large size and was considered to obstruct the pelvis inlet (Figure 5A,B). Based on these findings, a diagnosis of obstructive dystocia was made, and a transplastron coeliotomy was performed, taking into account the diagnosis of obstructive dystocia, the large number of eggs, as well as the abnormal appearance of several eggs and the variable size of the eggs.

Following bilateral salpingotomy, 3 eggs were removed from each oviduct. The most caudal egg needed to be moved in retrograde using gentle digital manipulation via the cloaca before it could be removed through the left salpingotomy incision. Next, bilateral oophorectomy was performed. The day following surgery, the turtle showed good appetite, and buoyancy was restored one week after the surgery. During an 18-month follow-up period, an increased food intake was noticed according to the owner which was reflected by a substantial gain in body weight to 1.35 kg. The transplastronectomy segment was removed as a bony sequestrum after 6.5 months, and bony growth could be noticed below.

### 3.6. Case 6

A 51-year-old female Hermann’s tortoise (*Testudo hermanni*) was presented because of chronic egg binding. Based on a ventrodorsal radiograph, the local vet diagnosed the presence of a presumed egg with an abnormal shape 2 years prior to initial presentation. Treatment with oxytocin by the local vet did not result in oviposition, and the owner declined further treatment and did not seek further veterinary advice as the tortoise did not show clinical signs at that time. Two years after the initial diagnosis, the animal was irresponsive and anorectic immediately after awakening from hibernation and was presented 3 weeks later. Based on the clinical examination as well as the biochemistry test results and hematological evaluation, severe dehydration, moderate hypercalcemia, and a marked rise in creatinine kinase, in addition to a relatively low PCV (17%) and mild heterophilia were demonstrated. A radiographic examination revealed a large radiopaque structure in the left caudal coelomic cavity (Figure 6A). The size, shape, and position of the structure were identical to the radiographic findings of the local vet.

Following 5 days of rehydration and nutritional therapy, the tortoise showed a marked improvement of the general condition, and a transplastron coeliotomy was performed to remove the retained egg from the left oviduct through a salpingotomy. The latter surgical approach was preferred, taking into account the large size of the retained structure and the chronicity of the condition. The cranial part of the oviduct seemed to be invaginated into this opening and could be reposed using digital manipulation (Figure 6B). The structure seemed to consist of a severely thickened eggshell with a central lumen (Figure 6C). Next, a bilateral oophorectomy was performed. The tortoise showed an obvious improvement in general condition and a restored appetite one week after surgery, respectively. During an 8-month follow-up period, the tortoise remained without clinical signs.

### 3.7. Case 7

A 5-year-old female Hermann’s tortoise was presented due to showing anorexia for one week, and apathy as well as open-mouth breathing for two days. Prior to showing anorexia, the tortoise had shown overactive and nesting behavior for 5 days. The tortoise showed prominent flattening and pyramiding of the carapace and decreased mineralization of the shell as well as bilateral distention of the prefemoral fossae. Based on an ultrasound, an overfilled urinary bladder was noticed as well as well-mineralized eggs that were considered to be present within the urinary bladder, and a percutaneous cystocentesis was performed. As the vertical distance between the caudal edge of the carapace and plastron was considered to be abnormally limited, it was presumed that the eggs had entered the urinary bladder from the cloacal vestibule through the urodeum and urethral opening as they could not pass through the vent. A radiographic examination was advised but declined by the owner. Taking into account the abnormal development of the tortoise as well as the presence of 4 eggs within the urinary bladder, a transplastron coeliotomy was performed, and following entry of the coelom, the presence of the eggs within the urinary bladder was confirmed. Next, a cystotomy was performed as previously described by Divers [13] to remove the eggs after intraoperative cystocentesis followed by bilateral oophorectomy.

Recovery from the surgery was smooth, and following a week of supportive care and force-feeding, the appetite was restored. Both management and nutritional advice were provided to the owner. During a 6-month follow-up period, the tortoise appeared to show a normal condition.

## 4. Discussion

The cases in the present report illustrate the successful use of endoscope-assisted prefemoral surgery as well as transplastron coeliotomy for various reproductive disorders in several chelonian species. Although most signs at initial presentation could be related to egg retention in 6 out of 7 cases and chronic retention was demonstrated in 3 cases, primary conditions such as neoplastic disease as well as secondary complications, such as salpingitis, ectopic eggs in the coelomic cavity or urinary bladder, and coelomitis need to be considered in cases with obstructive or nonobstructive dystocia [1,2,14].

Although transplastron coeliotomy is considered as an invasive procedure, it has been documented as an effective and safe procedure for the treatment of reproductive and other disorders (e.g., cystic calculus removal) in various chelonian species [8]. In the present report, four out of seven cases required the use of transplastron coeliotomy. The use of this technique versus an endoscopy-assisted prefemoral approach should be well-considered, and in chelonians this choice is primarily dictated by the involved species. In chelonian species with nonrestrictive prefemoral fossae (e.g., *C. serpentina, Sternotherus* spp., *Terrapene* spp., softshell turtles) [15], the endoscope-assisted surgery of reproductive disorders is often possible. Depending on the size of the animal, the size and number of retained eggs, or other pathology of the reproductive tract as well as complicating factors, a uni- or bilateral approach can be used [6]. As previously documented, endoscope-assisted oophorectomy usually can be performed through a single fossa in *Trachemys* sp. [6,10], as demonstrated for the cases involving terrapins as well as the snapping turtle in the present study. In the terrapins with the ovarian carcinoma (case 1) and the obstructive dystocia (case 5), a prefemoral approach would not have allowed successful treatment, while in the yellow-bellied slider (case 2) and the black marsh turtle (case 4), the prefemoral approach was adequate and a single point of entry proved to be sufficient for the removal of retained eggs and performing of a bilateral oophorectomy in the slider. In the latter case, however, intraoperative ovocentesis was necessary in order to remove multiple eggs from the coelomic cavity.

Taking into account the signalment of the described cases, it is noteworthy that 6 out of 7 cases had an average to relatively old age [15]. Several of the described cases had successfully produced clutches of eggs prior to developing dystocia and other reproductive disorders. Although several reproductive disorders in chelonians may be expected to be seen more frequently in aged animals (e.g., neoplastic disease) and solitary females [2], we do not consider the age as a fundamental risk factor in the development of reproductive disorders, and five out of seven cases in the present study were co-housed with a conspecific male.

Specifically, nutritional secondary hyperparathyroidism (NSHP) and other disorders that cause calcium deficiency are considered to predispose the development of dystocia and PFS in reptiles [2]. In the present study, mild to severe signs of skeletal demineralization were demonstrated in the black marsh turtle (case 4) and the Hermann’s tortoises (cases 6 and 7). Especially in case 7, the retrograde migration of eggs from the cloaca to the urinary bladder through the urodeum was undoubtedly related to developmental abnormalities of the shell that were primarily related to NSHP. Although relatively uncommon, the authors observed several cases of ectopic eggs in the coelomic cavity in saurian and chelonian species as seen in the yellow-bellied slider (case 2) and in a Hermann’s tortoise (case 7). In most cases, the repeated administration of usually high doses of oxytocin seemed to have a causal relationship with this pathology, especially in cases where obstructive dystocia was not recognized prior to the start of the treatment [14,16].

Several cases in the present study illustrate that chronic egg retention can be left unnoticed during long periods of time until secondary complications occur or the primary cause leads to overt signs. Radiographic signs of chronic egg retention may include the excessive mineralization of the eggshells that sometimes develops into a lamellar appearance after a prolonged period of time as observed in several cases in this study. In addition, abnormally shaped or collapsed eggs and the rupture of the eggshell with the subsequent development of salpingitis as observed in the black marsh turtle (case 4) constitute other pathologies that can be related to chronic egg binding.

While postovulatory dystocia, especially obstructive cases, can be considered as one of the most common reproductive disorders that requires surgical treatment, the prevalence of true PFS in chelonians seems to be low, and to the authors’ knowledge there are no literature data that unambiguously demonstrate cases of PFS in chelonians. Although other disorders, such as ovarian and oviductal neoplasia as described in the first case of this study, are also relatively rare in chelonians [17], oophoritis and salpingitis as diagnosed in the snapping turtle (case 3) and the black marsh turtle (case 4), respectively, may be underdiagnosed in chelonians in comparison to other reptile taxa. This may be largely attributed to the chelonian shell that hampers easy visualization and sampling of the reproductive tract.

## 5. Conclusions

There are various reproductive disorders in chelonians that necessitate the use of surgical procedures. We consider the series of disorders described in the present report as representative of the reproductive conditions in chelonians that may be encountered in veterinary practice and that require advanced surgical treatment. The findings of this study may aid the practitioner in establishing an etiological diagnosis, differentiating between conditions that need a conservative or surgical approach, and selecting the most suitable surgical treatment for these and comparable reproductive disorders. Although restrictions may be encountered, the endoscope-assisted prefemoral approach is the least invasive, and thus preferred, technique instead of the ventral transplastron coeliotomy. In conclusion, the therapeutic efficacy for the described cases largely relied on the choice of minimally invasive endoscopic-assisted surgery versus transplastron coeliotomy that was primarily dictated by the involved species, etiology, and associated pathology.

## Figures and Tables

**Figure 1 animals-12-03439-f001:**
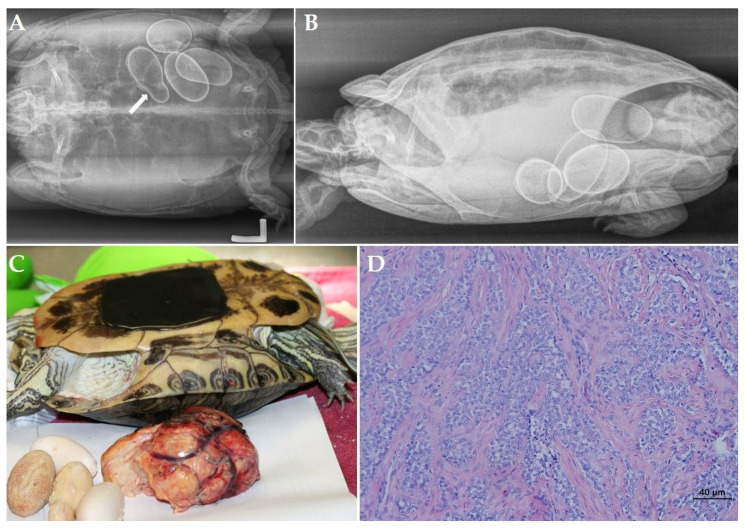
Chronic egg binding associated with a scirrhous ovarian carcinoma in a red-bellied slider (*Trachemys cripta elegans*). (**A**) Dorsoventral radiographic projection: 5 eggs located at the right side of the caudal coelomic cavity. An irregular shape is noted in 3 eggs and all eggs show abnormally thickened shells with a lamellar appearance in the most cranial egg. One egg has an unusual small size and is fused with one of the other eggs (arrow). (**B**) Generalized soft tissue opacity of the coelomic cavity with obvious compression of the lung field on the left laterolateral projection. (**C**) Postoperative view after removal of the ovarian neoplasia and retained eggs. (**D**) Histological section of a scirrhous ovarian carcinoma (Hematoxylin and eosin stain) composed of a cell-rich tissue that is growing infiltrative in dense collagen stroma. Neoplastic cells are organized in islets to multi-layered tubules.

**Figure 2 animals-12-03439-f002:**
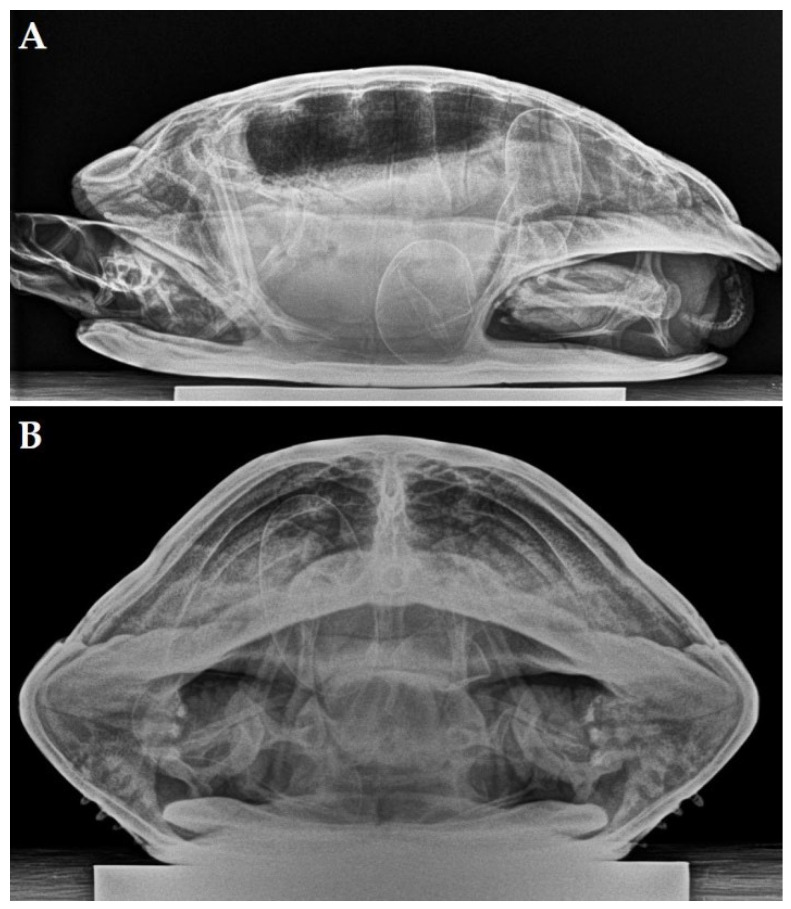
Laterolateral (**A**) and craniocaudal (**B**) radiograph (horizontal beam) of a yellow-bellied slider (*Trachemys scripta scripta*) revealing the presence of 4 ectopic eggs with 3 eggs that have a well-mineralized but thin shell and an abnormal shape. The fourth egg has an abnormally small size. The long axis of the most caudal, large egg showed a vertical position and is located in the right caudal coelomic cavity cranial to the pelvis inlet.

**Figure 3 animals-12-03439-f003:**
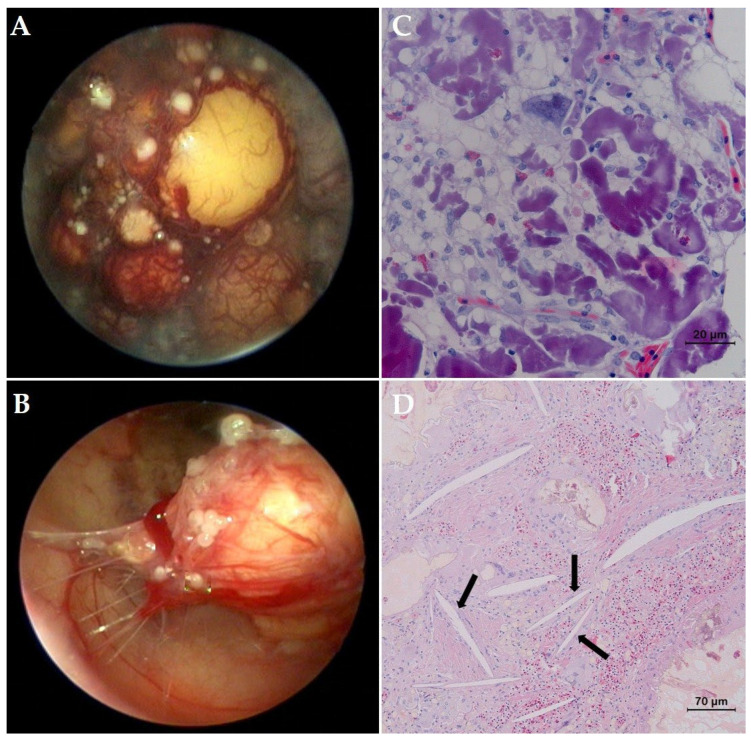
(**A**) Left prefemoral coelioscopic view of the left ovary in a common snapping turtle (*Chelydra serpentina*) with oophoritis. Previtellogenic and a large number of excessively vascularized early vitellogenic follicles can be observed. (**B**) Several follicles show adhesions to the coelomic membrane. (**C**) Histological section of the ovarian tissue (Hematoxylin and eosin stain). Note the presence of multiple follicles surrounded by large numbers of foamy macrophages and giant cells. (**D**) In the ovarian parenchyma, the multifocal presence of cholesterol crystals (arrows) surrounded by macrophages and giant cells can be observed.

**Figure 4 animals-12-03439-f004:**
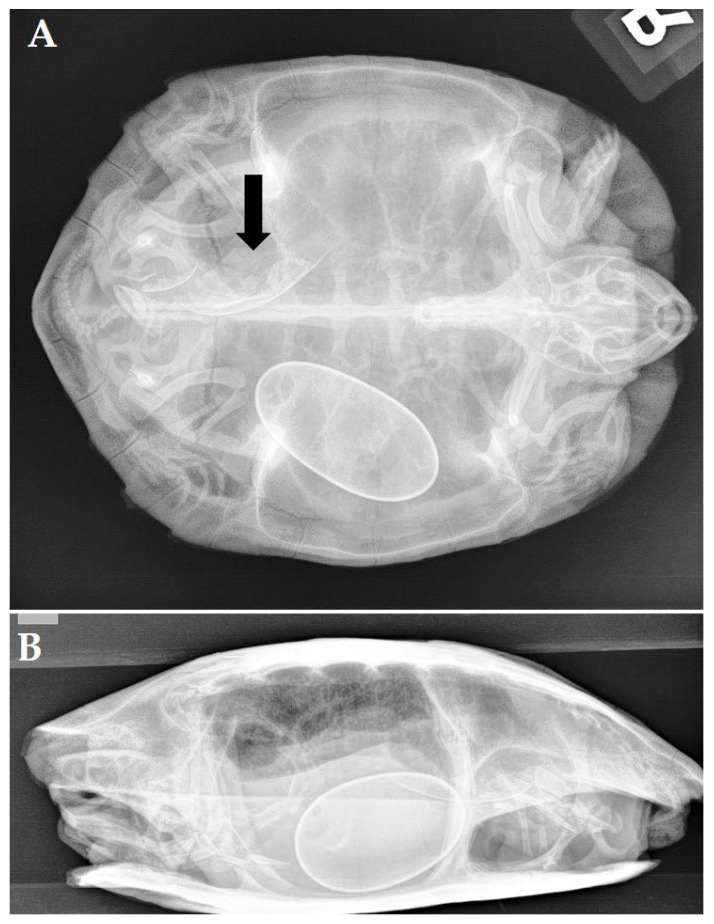
Ventrodorsal (**A**) and laterolateral (**B**) radiograph of a black marsh turtle (*Siebenrockiella crassicollis*) showing mild generalized decreased mineralization of the skeleton and a large, excessively calcified egg in the left caudal coelomic cavity as well as the remnants of an eggshell (arrow) in the right coelomic cavity.

**Figure 5 animals-12-03439-f005:**
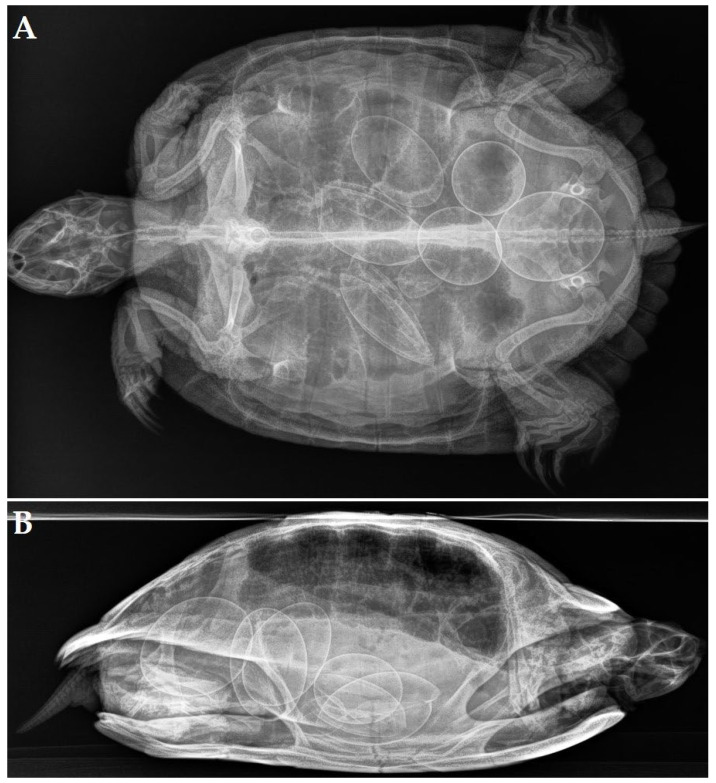
Ventrodorsal (**A**) and laterolateral (**B**) radiograph of a red-eared slider (*Trachemys scripta elegans*) with obstructive dystocia revealing the presence of 7 eggs with a well-mineralized shell. The 2 most cranial eggs with a normal size as well as an excessively small egg are collapsed, and the most caudal egg has a relatively large size and obstructs the pelvis inlet.

**Figure 6 animals-12-03439-f006:**
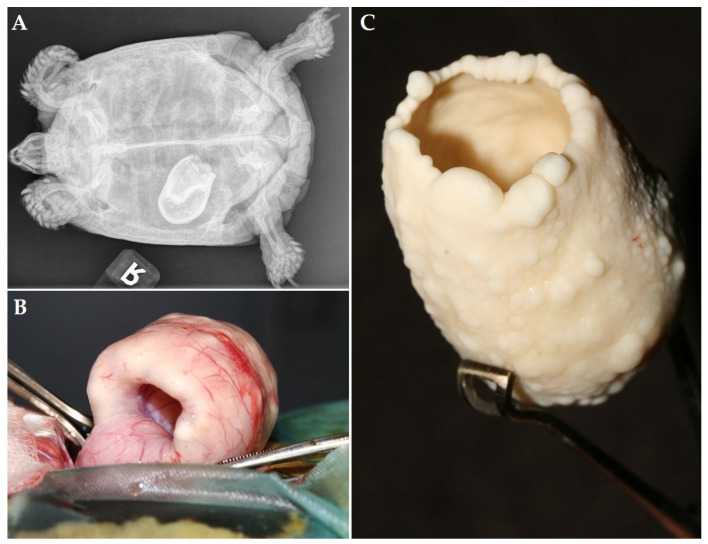
(**A**) Dorsoventral radiograph of a Hermann’s tortoise (*Testudo hermanni*) showing a large, retained egg with an abnormal shape in the left caudal coelomic cavity. (**B**) Intraoperative view during transplastron coeliotomy exteriorization of the oviduct containing the retained egg. (**C**) The egg after surgical removal from the oviduct.

**Table 1 animals-12-03439-t001:** Overview of chelonian cases with reproductive disorders that were treated with transplastron coeliotomy (TC) or endoscope-assisted coeliotomy. Tx: treatment; Dx: diagnosis; CT: cystotomy; OE: oophorectomy; EAO: endoscope-assisted oophorectomy; ST: salpingotomy.

Case nr.	Species	Age, Body Weight	Management and Nutrition	Clinical Signs	Dx	Tx	Follow-Up Period
1	Red-eared slider (*Trachemys scripta elegans*)	29 years, 0.85 kg	Housing: glass tank of 200 L. Heat bulb installed in the dry part (35 × 50 cm) of the enclosure creating a local hotspot of 28 °C. UV irradiation not provided. Diet: cat food pellets on a daily basis. Co-housed with a female (20 yrs) and male (18 yrs) red-eared slider.	Apathy, anorexia	Scirrhous ovarian carcinoma, chronic egg binding	TC, OE	24 months
2	Yellow-bellied slider (*Trachemys scripta scripta*)	14 years, 1.185 kg	Housing: aquarium containing 120 L of water and dry area (50 × 50 cm) covered with sand as a substrate. Ultraviolet irradiation provided with an average temperature at the basking spot of 27 °C. Diet: commercial cat food pellets combined and gammarus on a daily basis. Co-housed with a female yellow-bellied slider of the same age.	Apathy, anorexia	Ectopic eggs, coelomitis, chronic egg binding	EAO	48 months
3	Common snapping turtle (*Chelydra serpentina*)	32 years, 11.2 kg	Housing: outside pond (6 × 3 × 1.2 m) where the animals hibernated from early September to late April. Diet: adult mice and liver twice a week. Co-housed with a male snapping turtle of the same age.	Post-hibernation anorexia	Granulomatous oophoritis	EAO	18 months
4	Black marsh turtle (*Siebenrockiella crassicollis*)	19 years, 0.34 kg	Housing: inside pond of 80 L and a dry area of 80 × 80 cm with cocopeat as substrate. UV irradiation provided with hotspot of 29 °C. Diet: commercial turtle pellets, liver once every 2 days. Co-housed with two females and a male black marsh turtle of the same age. UV irradiation was provided in the dry part and a water part of 80 L was present.	Apathy, anorexia	Bacterial salpingitis	ST	12 months
5	Red-eared slider (*Trachemys scripta elegans*)	22 years, 1.1 kg	Glass tank of 150 L with a dry part (40 × 30 cm) provided with local hot spot of on average 27 °C created by an UV irradiation heat bulb. Diet: commercial turtle pellets on a daily basis. Co-housed with a male and female red-eared slider of the same age.	Tenesmus, buoyancy disorder	Obstructive dystocia	TC, ST, OE	18 months
6	Hermann’s tortoise (*Testudo hermanni*)	51 years, 0.98 kg	Housing: outside enclosure of 6 × 3 m with glass house of 80 × 50 × 80 cm. Individually housed. Diet: herbs and vegetables on a daily basis.	Apathy, anorexia, dehydration	Chronic egg binding, oviductal invagination	TC, ST, OE	8 months
7	Hermann’s tortoise (*Testudo hermanni*)	5 years, 0.630 kg	Housing: terrarium of 150 × 60 × 40 cm with wood chips as a substrate. A heat lamp creating a local hotspot of 27 °C on average. Neither UV irradiation, calcium, nor vitamin supplementation were provided. Co-housed with a male Hermann’s tortoise of same age. Diet: vegetables on a daily basis.	Anorexia, apathy, dyspnea	Dystocia, ectopic eggs within urinary bladder	TC, CT, OE	6 months

## Data Availability

The data presented in this study are available on request from the corresponding author.

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
