# Peer review of "The Use of Prefemoral Endoscope-Assisted Surgery and Transplastron Coeliotomy in Chelonian Reproductive Disorders"

_animals, 2022, doi:10.3390/ani12233439_

Round 1

Reviewer 1 Report

Overall nice case reports, soundly documented and preseted. Nevertheless the authors need to focus more on the evaluation process of the surgery. For veterinarians with a lot of reptile cases it might be easy to understand what kind of surgical approach to use. For veterinarians with few reptile cases it might be better to describe in detail on what terms the authors choose the surgical approach for which case.

Added comments:

1.The main question should be the evaluation process of deciding the surgical approach transplatron or prefemoral. Therefore the evaluation process could be more detailed by the author’s description. 

2.Both surgical approaches are well known an researched. Nevertheless evaluation of decision based on case reports is helpful for practitioners. 

3.The paper should focus a little more on the evaluation process of the surgery. This could help practitioners with their own cases.

4.The tables and figures give a good overview over the cases. They could be specifically used for the evaluation process. For example: is there a species dependent cause for the chosen surgical approach? Are there species specific difficulties for the surgical approach?  Are there case related issues for uni or bilateral approach? 

Reviewer 2 Report

Dear authors, it is good idea to describe advantage as well as disadvantage of two surgical approaches to chelonian reproductive disorders. But you should improve your manuscript:

Title - your title The use of  prefemoral....... versus transplaston ... seems that you compare two methods on similar conditions. But you did not. So, please, would you be ready to improve the title?

Abstract -  a big part of the abstract is more introduction than abstract (lines 18 - 23 could be cancelled)

Materials and Methods - (did you really use the meloxicam in such low doses 0.3 mg/kg?);  Table 1 - the method EAO is not written in case nr.4; Case 2 - is more fit for the transplastral surgery - orientation of endoscope when looking for free eggs in coelom is not easy when follicles are present; Case 3 - lines 203-205 and Case 4 - lines 243 - 250 - you can describe in more details the method please. Because two possibilities exist - endoscope-assisted surgery (endoscope is used only for orientation inside the coelom and for grasping mesovarium x endsoscope is used for minimal invasive surgery inside the coelom) - please describe which of those methods were used please.

Page 13 - PFS? do you mean POFS (pre ovulatory stasis syndrome)?
